# Validation of MRI-Based Models to Predict MGMT Promoter Methylation in Gliomas: BraTS 2021 Radiogenomics Challenge

**DOI:** 10.3390/cancers14194827

**Published:** 2022-10-03

**Authors:** Byung-Hoon Kim, Hyeonhoon Lee, Kyu Sung Choi, Ju Gang Nam, Chul-Kee Park, Sung-Hye Park, Jin Wook Chung, Seung Hong Choi

**Affiliations:** 1Institute of Behavioral Sciences in Medicine, College of Medicine, Yonsei University, Seoul 03722, Korea; 2Department of Psychiatry, College of Medicine, Yonsei University, Seoul 03722, Korea; 3Department of Anesthesiology and Pain Medicine, Seoul National University Hospital, Seoul 03080, Korea; 4Biomedical Research Institute, Seoul National University Hospital, Seoul 03080, Korea; 5Artificial Intelligence Collaborative Network, Department of Radiology, Seoul National University Hospital, Seoul 03080, Korea; 6Department of Radiology, Seoul National University Hospital, Seoul 03080, Korea; 7Department of Neurosurgery, Seoul National University Hospital, Seoul 03080, Korea; 8Department of Pathology, Seoul National University Hospital, Seoul 03080, Korea; 9Institute of Innovate Biomedical Technology, Seoul National University Hospital, Seoul 03080, Korea; 10Center for Nanoparticle Research, Institute for Basic Science (IBS), Seoul 08826, Korea

**Keywords:** gliomas, neural network, MRI, O6-methylguanine-DNA methyl transferase, radiogenomics

## Abstract

**Simple Summary:**

O6-methylguanine-DNA methyl transferase (MGMT) methylation in glioblastoma is an important prognostic and predictive factor that requires an invasive surgical procedure for identification. In several recent studies, MGMT methylation prediction models were developed using MR images, and good diagnostic performance was achieved, which seems to indicate a promising future for radiogenomics. However, the diagnostic performance was not reproducible for numerous research teams when using a larger dataset in the RSNA-MICCAI Brain Tumor Radiogenomic Classification 2021 challenge. To our knowledge, there has been no study regarding the external validation of MGMT prediction models using large-scale multicenter datasets. We tested recent CNN architectures via extensive experiments to investigate whether MGMT methylation in gliomas can be predicted using MRI. With unexpected negative results, approximately 80% of the developed models showed no significant difference with the chance level of 50% in terms of external validation accuracy. In conclusion, MGMT methylation status of gliomas may not be predictable with preoperative MRI, even using deep learning.

**Abstract:**

O6-methylguanine-DNA methyl transferase (MGMT) methylation prediction models were developed using only small datasets without proper external validation and achieved good diagnostic performance, which seems to indicate a promising future for radiogenomics. However, the diagnostic performance was not reproducible for numerous research teams when using a larger dataset in the RSNA-MICCAI Brain Tumor Radiogenomic Classification 2021 challenge. To our knowledge, there has been no study regarding the external validation of MGMT prediction models using large-scale multicenter datasets. We tested recent CNN architectures via extensive experiments to investigate whether MGMT methylation in gliomas can be predicted using MR images. Specifically, prediction models were developed and validated with different training datasets: (1) the merged (SNUH + BraTS) (*n* = 985); (2) SNUH (*n* = 400); and (3) BraTS datasets (*n* = 585). A total of 420 training and validation experiments were performed on combinations of datasets, convolutional neural network (CNN) architectures, MRI sequences, and random seed numbers. The first-place solution of the RSNA-MICCAI radiogenomic challenge was also validated using the external test set (SNUH). For model evaluation, the area under the receiver operating characteristic curve (AUROC), accuracy, precision, and recall were obtained. With unexpected negative results, 80.2% (337/420) and 60.0% (252/420) of the 420 developed models showed no significant difference with a chance level of 50% in terms of test accuracy and test AUROC, respectively. The test AUROC and accuracy of the first-place solution of the BraTS 2021 challenge were 56.2% and 54.8%, respectively, as validated on the SNUH dataset. In conclusion, MGMT methylation status of gliomas may not be predictable with preoperative MR images even using deep learning.

## 1. Introduction

Glioblastoma is the most common primary central nervous system malignancy, with a devastating median survival of 14.6 months even after operation followed by concurrent chemoradiation therapy (CCRT) with adjuvant temozolomide (TMZ) [1]. Stupp et al. showed that patients with O6-methylguanine-DNA methyl transferase (MGMT) silencing by methylation have a survival benefit when treated with temozolomide. Their median survival was 21.7 months compared with 12.7 months for those who were not treated [2], even with long-term follow-up [3]. MGMT antagonizes the lethal effect of alkylating agents by removing the alkyl adduct at the O6 position of guanine at the DNA level, decreasing the cytotoxic effect of TMZ, which is an alkylating agent. During the repair process, the methylation of MGMT induces irreversible inhibition of MGMT function, leading to inadequate DNA repair [4]. Thus, MGMT methylation is a crucial treatment-predictive factor in glioblastoma, which increases the chemosensitivity of TMZ [4,5]. Moreover, MGMT methylation is prognostic, and showed longer median overall survival in the methylated group than in the unmethylated group in a randomized controlled study. During follow-ups, pseudoprogression after CCRT was more common in tumors with MGMT promoter methylation in GBM, and methylation of the MGMT promoter should be considered when interpreting follow-up MRI.

In some previous works, it was reported that glioblastoma with MGMT methylation showed mass-like edema with nodular enhancement, whereas glioblastoma without MGMT methylation showed infiltrative edema with thick enhancement [6]. Based on these radiological features of conventional MRI, for over a decade, many researchers developed models to predict MGMT, including radiomics approaches using high-throughput quantitative imaging features [7]. However, in most of these studies, only a small dataset was used, or external validation was not performed, which is insufficient to report generalizability. Moreover, performance has also been inconsistent. Han et al. [8] demonstrated a validation accuracy of 67%, and Wei et al. showed a validation accuracy of 77% (*n* = 31). Finally, the RSNA-MICCAI Brain Tumor Segmentation (BraTS) 2021 Radiogenomic Classification challenge, or task 2 of the BraTS 2021 challenge, was held [9], and the first place solution showed a test AUROC of 0.62, which showed a large discrepancy in performance when compared with another previous study that reported a three-fold cross-validation accuracy of 94.73% [8,10,11,12], even using a subset of the same TCIA dataset (*n* = 247).

The generalization of models to different datasets is one of the biggest challenges for the application of artificial intelligence techniques in clinical radiology practice [13]. However, to our knowledge, there have been few previous reports regarding large-scale datasets of MGMT methylation as well as external validation. The aim of our study was to determine whether mpMRI can be used to predict MGMT promoter methylation status using the largest dataset (*n* = 985). We performed extensive validation of CNN-based prediction models with large-scale public and independent private datasets.

## 2. Methods and Materials

### 2.1. Datasets

The Institutional Review Board of Seoul National University Hospital (SNUH) approved this retrospective study with a waiver of informed consent, and this study was carried out in accordance with the Declaration of Helsinki. From November 2014 to January 2019, 434 patients over 18 years old were consecutively and retrospectively enrolled in this study according to the following inclusion criteria: (1) histopathologic diagnostic confirmation of diffuse gliomas (i.e., WHO grade II-IV astrocytoma, oligodendroglioma, and glioblastoma) based on 2016 WHO CNS tumor classification at the initial operation; (2) treatment-naïve MRI, and (3) underwent standard treatment, including surgery or stereotactic biopsy. A total of 34 patients were excluded because of the following: (1) missing sequences (*n* = 7); (2) suboptimal image quality and consequent poor coregistration (*n* = 8), and (3) unavailable initial MGMT promoter methylation status (*n* = 19) (Figure 1). Finally, a total of 400 patients who underwent all four conventional MRI scans were enrolled in the study. Because the WHO classification system was changed in 2021, which was the time after the pathologic diagnosis was made, the pathologic diagnosis of the tumors was re-classified after enrollment. Detailed information regarding tissue diagnosis and genetic analysis is provided in the Appendix A.

To alleviate the small training dataset size, we combined the dataset from our institution, SNUH, with the BraTS 2021 training dataset (*n* = 585), which is a multicentered dataset from the RSNA-MICCAI Brain Tumor Radiogenomic Classification challenge (BraTS 2021 challenge) [9]. The BraTS dataset, one of the largest benchmarks of brain tumor, is a retrospectively collected dataset of brain tumor and mpMRI scans acquired from multiple institutions under standard clinical conditions. Inclusion criteria comprised pathologically confirmed diagnosis of diffuse gliomas and available MGMT promoter methylation status [9]. For both the BraTS and SNUH datasets, mpMRI sequences of the brain were obtained including fast spin-echo T1-weighted imaging (T1w), T2-weighted imaging (T2w), T2-weighted fluid-attenuated inversion recovery (FLAIR), and contrast-enhanced T1-weighted imaging (T1wCE). For the SNUH dataset, each T1wCE experiment was performed using a three-dimensional magnetization prepared rapid gradient echo (3D MPRAGE) sequence before and after the administration of gadobutrol (Gadovist; Bayer, Berlin, Germany; at a dose of 0.1 mmol/kg of body weight) in the enrolled patients. The MR scan parameters are detailed in Appendix A.

### 2.2. Data Preprocessing and Model Implementation

All datasets were preprocessed using skull-stripping, coregistration, and intensity rescaling, in an identical manner (Appendix A). For the model architectures, we utilized four representative convolutional neural network (CNN) architectures: EfficientNet [14], squeeze-and-excitation networks [15], aggregated residual transformations [16] (i.e., SEResNet and SEResNeXt), and DenseNet [17], which are known to show robust performance to medical image classification tasks. The models were originally developed for 2D images; however, they were reconstructed as 3D CNNs to efficiently take and process the input 3D MR images. All the details on the data preprocessing and model implementation are provided in the Appendix A.

### 2.3. Experiments

To examine the effect of datasets on model performance, a total of three kinds of experiments were performed based on different datasets (d = 3). We (1) trained the model using the SNUH dataset and validated the model using the BraTS 2021 dataset (Experiment 1); (2) trained the model using the BraTS 2021 dataset and validated the model using the SNUH dataset (Experiment 2), and (3) trained and validated the model using the merged (SNUH + BraTS) dataset (Experiment 3). When using the merged dataset, we randomly split it into training, validation and test sets using an 8:1:1 ratio.

For each experiment, we validated (1) four representative convolutional neural network (CNN) architecture-based models (m = 4), and (2) seven combinations of sequences (s = 7) as input channels to examine the effect of model architectures and input MRI sequences on the model performance. Thus, a total of 28 combinations were validated in each experiment. Specifically, we tested (1) EfficientNetB0; (2) SEResNet50; (3) SEResNeXt50; and (4) DenseNet121. The model architecture is detailed in the Appendix A. For seven combinations of sequences, we chose the following sequences as input features, based on previous studies, concatenated to the channel axis: (1) FLAIR only; (2) T1w only; (3) T1wCE only; (4) T2w only; (5) FLAIR and T1wCE; (6) FLAIR, T1wCE, and T2w; and (7) FLAIR, T1wCE, T2w, and T1w, or all the sequences. Thus, a total of 420 (d × m × s × r) experiments were performed. To ensure the robustness of the results, each experiment was repeated five times using different seed numbers (i.e., 0, 42, 1234, 1000 and 9999) in each run. The area under the receiver operating characteristic curve (AUROC), accuracy, precision, and recall were assessed for model evaluation. The best models were selected based on the highest AUROC. In addition, the reproducibility of the BraTS 2021 challenge first and second place solutions were externally validated using the SNUH dataset (Experiment 4).

Statistical analysis is detailed in the Appendix A.

## 3. Results

### 3.1. Patient Characteristics

In the SNUH dataset (*n* = 400; 240 (60%) men, mean age, 52.3 ± 15.2 years old), age was not significantly different between methylation groups (*p* = 0.655) (Table 1). According to WHO CNS tumor classification 2021, patients with adult-type diffuse gliomas were enrolled in the study: astrocytoma, IDH-mutant (*n* = 63); oligodendroglioma, IDH-mutant, 1p/19q-codeleted (*n* = 33); and glioblastoma, IDH-wildtype (*n* = 304). MGMT methylation occurred in (1) 307 out of 585 patients (52.5%) in the BraTS dataset (*n* = 585); (2) 197 out of 400 patients (49.3%) in the SNUH dataset, and (3) 504 out of 985 patients (51.2%) in the merged dataset. The proportion of methylation was not significantly different between the two datasets (*p* = 0.319). Progression-free survival (PFS) was significantly longer (median, 396 days (95% confidence interval (CI), 328–526 days)) vs. 974 days (95% CI, 698–1302 days)) in the methylated group than in the unmethylated group (*p* < 0.0001) (Appendix A). Patients with methylated MGMT showed a hazard ratio (HR) of 0.50 (95% CI, 0.37–0.67). In subgroup analysis of glioblastoma, IDH-wildtype according to WHO CNS tumor classification 2021, PFS was significantly longer (median, 514 days (95% CI, 438–676 days) vs. median, 328 days (95% CI, 285–387 days)) in the methylated group than in the unmethylated group (*p* = 0.0001) (Appendix A), and patients with methylated MGMT showed a HR of 0.51 (95% CI, 0.36–0.72). However, PFS showed no difference between the MGMT methylated and unmethylated groups in the IDH-mutant subgroup: mean, 1949 (95% CI, 1722–2177) vs. 1650 (95% CI, 1263–2037) days (*p* = 0.871); HR, 0.91 (95% CI, 0.30–2.76). Detailed patient characteristics for the SNUH dataset are summarized in Table 1. Patient characteristics for the BraTS dataset are available by the BraTS 2021 challenge [9].

### 3.2. Model Performance

According to seven articles searched from PubMed, the mean dataset size was 155.4 ± 84.6 (range, 59–262), and the diagnostic performance was representing by an accuracy of 62–94.7% (Table 2). Specifically, the validation dataset showed variable sizes ranging from 20 to 82. Cross-validation was performed in two out of seven studies (3 and 10-fold) and externally validation was performed. Three out of seven models had 3D input. A deep learning approach was used in two out of seven studies (3D-DenseUNet and 2D-CNN with slice-direction RNN). All the studies used either T2w or T2 FLAIR images. All the studies used only part of all four MR sequences (i.e., 1–3 out of 4 sequences), except for one study.

In our experiments, only 43.8% (184/420) and 39.8% (167/420) of the 420 developed models showed better performance than chance level (50%) in terms of test accuracy and test AUROC, respectively. In the one proportion z-test, 80.2% (337/420) and 60.0% (252/420) of the developed models showed no significant difference with chance level (50%) in terms of test accuracy and test AUROC (*p* > 0.05), respectively.

In Experiment 1, the best neural network model among 140 models achieved the best AUROC, accuracy, precision, and recall of (1) 46.8% (mean, 46.4 ± 4.7%; range, 39.1–52.0%), 55.9% (mean, 55.9 ± 1.2%; range, 54.2–57.6%), 56.1% (mean, 55.9 ± 2.8%; range, 53.4–60.7%), and 74.2% (mean, 83.2 ± 18.7%; range, 54.8–100.0%) on the validation set (Table 3 and Figure 2a); and (2) 57.2% (mean, 51.6 ± 3.8%; range, 47.0–57.2%), 50.8% (mean, 49.8 ± 1.3%; range, 48.5–51.5%), 50.0% (mean, 49.0 ± 1.8%; range, 45.9–50.5%), and 96.4% (mean, 80.4 ± 31.5%; range, 25.9–100.0%) on the test set using the BraTS dataset for training. The best CNN architecture was EfficientNet-B0 using FLAIR-T1wCE-T2w-T1w (all four sequences) as input sequences (Table 4 and Figure 2b).

In Experiment 2, the best neural network model among 140 (m × s × r) models achieved AUROC, accuracy, precision, and recall of (1) 67.5% (mean, 57.8 ± 8.3%; range, 46.0–67.5%), 62.5% (mean, 55.5 ± 5.4%; range, 50.0–62.5%), 57.1% (mean, 36.6 ± 34.0%; range, 0.0–71.4%), and 100.0% (mean, 44.0 ± 49.9%; range, 0.0–100.0%) on the validation set (Table 3 and Figure 2a); and (2) 52.8% (mean, 51.5 ± 2.7%; range, 48.5–55.5%), 54.6% (mean, 50.4 ± 3.3%; range, 47.3–54.6%), 54.1% (mean, 32.6 ± 5.7%; range, 49.4–64.3%), and 90.2% (mean, 38.2 ± 43.7%; range, 0.0–90.2%) on the test set using the SNUH dataset for training/validation, and the BraTS dataset for testing, respectively. The best CNN architecture was SEResNet50 using T2w as the input sequence (Table 4 and Figure 2b).

In Experiment 3, the best neural network model among 140 (m × s × r) models achieved AUROC, accuracy, precision, and recall of (1) 48.8% (mean, 54.9 ± 5.4%; range, 48.8–63.1%), 55.1% (mean, 57.1 ± 3.1%; range, 53.9–61.8%), 53.8% (mean, 61.6 ± 10.3%; range, 53.2–75.0%), and 91.3% (mean, 68.7 ± 33.4%; range, 26.1–97.8%) on the validation set (Table 3 and Figure 2a); and (2) 64.5% (mean, 51.7 ± 7.7%; range, 45.9–64.5%), 55.6% (mean, 51.7 ± 7.7%; range, 45.9–64.5%), 54.1% (mean, 54.6 ± 5.7%; range, 49.4–64.3%), and 90.2% (mean, 62.0 ± 36.5%; range, 17.6–94.1%) on the test set using the merged dataset to randomly split into training, validation and test sets, respectively. The best CNN architecture was SEResNext50 using FLAIR-T1wCE as input sequences (Table 4 and Figure 2b).

In Experiment 4 or the validation on the BraTS 2021 challenge, the first and second place solutions achieved AUROC, accuracy, precision, and recall of (1) 56.2, 54.8, 53.6, and 59.9% using 3D-ResNet and T1wCE only as the input sequence; and (2) 51.2, 51.3, 51.4, and 18.3% using EfficientNetB0-LSTM and FLAIR-T1wCE-T2w-T1w (all the four sequences) as input sequences, respectively, as validated on the SNUH dataset (Table 4). We could not validate the third place solution, which was an ensemble model with different combinations of input MRI sequences, based on EfficientNetB3.

Comparing the test AUROC among different trained datasets, the best models trained/validated on the BraTS dataset and tested on the SNUH dataset showed better validation AUROC and accuracy; however, there were no significant differences (all *p* > 0.05 in pairwise comparison) (Figure 2). The best models trained/validated on the merged dataset and tested on the merged dataset showed better test AUROC and accuracy, which was also not significant (Figure 2). Comparing the performance among different combinations of MRI sequences, combinations including the T2 FLAIR sequence showed better test AUROC and accuracy than others; however, there was no significant difference (all *p* > 0.05 in pairwise comparison) (Figure 2). The best models among the 420 models regarding AUROC are listed in Table 3 and Table 4 to show the model performance. The distribution of data points according to the MGMT labels, specifically the probability scores predicted from the best models of each experiment, are provided in a swarm plot (Figure 3). Full metrics are detailed in the Appendix A. Only AUROCs over 0.5 are listed because an AUROC under 0.5 indicates that the model performance is worse than that of the chance level.

## 4. Discussion

MGMT methylation is an important predictive factor in glioblastoma, increasing the chemosensitivity of TMZ [4,5], playing a role as a prognostic factor and showing longer median overall survival than that of unmethylated groups [21]. As a result, previous attempts have been made to predict MGMT methylation in gliomas using MRI in various approaches. However, the diagnostic performances of these approaches were inconsistent, yielding large variance [8,10,11,12,18,19,20]. To our knowledge, this study is the first large-scale multicenter validation study to investigate the generalizability of MGMT methylation prediction using MRI in a deep learning approach.

In conventional MRI findings, nodular enhancement, ill-defined enhancing tumor margins, and mass-like edema have been reported to be associated with MGMT methylation. In contrast, infiltrative edema and thick enhancement have been reported to be associated with unmethylated MGMT status [6,18,22]. Based on these MRI findings, several prediction models using radiomics, textural analysis, and neural networks have been developed in the past decade. However, in most of the previous studies, only small-sized datasets (range, 59~264 cases) were used, and external validation using an independent dataset was not performed [8,10,11,12], discouraging the deployment of the developed models in clinical practice. Specifically, large variance was observed with respect to diagnostic performance, with a range of 62–96% accuracy (Table 2). While most studies showed validation performance of less than 70% accuracy [8], one study showed approximately 95% accuracy [12]. Although the diagnostic performance is variable and the dataset size is too small (smaller than *n* = 262 patients; because of the rarity of the disease), which increases the risk of overfitting, external validation has rarely been performed. Moreover, because the prediction models have variable validation dataset sizes, their predictive performance are highly dependent on the variable validation method used (i.e., cross-validation, random split internal validation, or external validation)3, 4 [23,24]. Thus, comparing the performance of models based on metric figures without specifying them may yield unreliable results. In this context, new reporting standards, such as TRIPOD [13], are required and should be validated carefully by a more thorough external dataset, especially when the training dataset is small (*n* < 1000) [25].

We developed and validated MGMT methylation prediction models in conjunction with the BraTS 2021 dataset, comprising the largest dataset (*n* = 985) of nearly one thousand patients, and meticulously tested various combinations of models, input sequences, and datasets. The seed number for randomness was also changed and compared because changing the seed number should not significantly affect model performance. As a result, in more than 80% and 60% of the developed models, we did not see a significant difference in terms of test AUROC and accuracy (*p* > 0.05), respectively, compared with the chance level of 50% in the one proportion z-test for the prediction of MGMT methylation (Figure 2). In addition, the first place solution of the BraTS challenge was the best model among all experiments and showed values of 56.2, 54.8, 53.6, and 59.9% for AUROC, accuracy, precision, and recall, respectively (Table 4). However, there were no significant differences in diagnostic performance among the different prediction models from Experiments 1–4 (all *p* > 0.05). In summary, comparing the test AUROC among different trained datasets, different CNN architectures, and different combinations of MRI sequences, there were no significant differences in diagnostic performances in the test sets (all *p* > 0.05). Interestingly, the best models in Experiment 1 (i.e., training/validation on the BraTS dataset and testing on the SNUH dataset) showed high mean test recall (mean, 80.4 ± 31.5%) using EfficientNet-B0 and all the MRI sequences. However, they also showed large variance in test recall, ranging from 25.9% to 100.0%, which indicates unstable performance of the prediction models. Finally, the validation metrics were generally better than the test metrics (Figure 2a,b) because model training was stopped early according to the high validation accuracy.

In the BraTS 2021 challenge, given the large amount of data (*n* = 585) with multiparametric inputs, none of the participants among 1555 worldwide teams, including the first place team (AUROC, 0.62), could discover reliable MR imaging features that correlate with MGMT methylation in gliomas. Further analysis of the BraTS 2021 challenge is detailed in the Appendix A. The test AUROC and accuracy of the first place solution of the BraTS 2021 challenge were 56.2% and 54.8%, respectively, when externally validated on the SNUH dataset, which is not sufficient performance for clinical application. Even combining an additional dataset from our institution, which comprises the largest (*n* = 985) dataset in total, the best model among the 420 developed models showed an AUROC and accuracy of 64.5% (mean, 51.7 ± 7.7%) and 55.6% (mean, 51.9 ± 3.4%), respectively, on the test set using the merged dataset. In our study, it is not yet clear what causes the poor performance. This may be caused by (1) overfitting due to the small dataset size, and (2) different data distributions from different hospitals. However, considering that the prediction of IDH mutation in gliomas has shown better performance, achieving 87–92% accuracy in distinguishing IDH mutant gliomas from IDH wild-type tumors using only conventional MRI [26,27], although this classification task is a more difficult task than MGMT methylation prediction because of the class imbalance of IDH mutation, it might be more plausible that MGMT mutations may not actually be reflected by a noticeable change in mpMRI.

Conflicting results have also been reported, in that none of the conventional MRI features showed significant differences between the two groups with or without MGMT methylation [23,28,29,30,31,32,33]. Moreover, as frequent epigenetic changes, MGMT methylation status changed by approximately 15% over the course of treatment 35. This may support the idea that methylation of the MGMT promotor is not well reflected by conventional mpMRI, and conventional imaging findings may be nonspecific except for less edema. Interestingly, methylation of the MGMT promotor has a low extent of edema, low apparent diffusion coefficient (ADC), and low relative cerebral blood volume (rCBV) [7].

The imaging phenotype of gliomas in multiparametric MRI is largely dependent on genetic features other than MGMT methylation, such as IDH mutation. There appear to be far more clinical and genetic features that more strongly determine the morphology of gliomas in conventional MRI, and the methylation status of the MGMT promotor is only one of the “weak” features in determining the imaging phenotype. Thus, incorporating other genotypic information as well as advanced MR sequences, such as diffusion-weighted and perfusion-weighted imaging, which can provide cellularity and angiogenesis for tumor characteristics in gliomas [31,34,35], may aid in the discovery of “strong” imaging features with higher correlations than conventional MR sequences to improve the predictive power of MGMT methylation [36]. In summary, using a larger dataset of only conventional MRI sequences may not significantly improve the diagnostic performance, which suggests that additional information is required for improvement, hopefully in a prospective study in the future.

There are several limitations to this study. (1) We cannot conclude that MGMT prediction is “impossible” using conventional mpMRI. However, it was not achievable with a dataset size of nearly 1000 via thorough experiments, including the external validation of the first place solution of the BraTS 2021 challenge. (2) The risk of overfitting may be alleviated by increasing the number of datasets (*n* = 985) to a relatively small dataset using modern “heavy” neural network architectures with a large number (i.e., tens of millions) of parameters, which can be used to extract good latent features compared with previous models. (3) The prediction model did not consider IDH status because the BraTS dataset did not have labels of IDH status for training. This warrants further investigation because IDH status is crucial in the new WHO CNS tumor classification 2021. For future studies, we can incorporate the IDH status to the prediction model for the MGMT methylation in diffuse gliomas because IDH-wildtype and IDH-mutant group shows different tumor biology, although they are classified as adult-type diffuse gliomas, which might lead to different diagnostic performance of MGMT methylation.

## 5. Conclusions

In conclusion, contrary to expectations, MGMT methylation cannot be predicted using only conventional structural MRI, even in a deep learning approach with a large-scale multicenter dataset. Although radiogenomics has the capacity to alter how patients with brain tumors are managed in the future, additional tumor characteristics from advanced MRI, such as cellularity and angiogenesis, should be pursued to improve the noninvasive diagnostic performance of MGMT methylation.

## Figures and Tables

**Figure 1 cancers-14-04827-f001:**
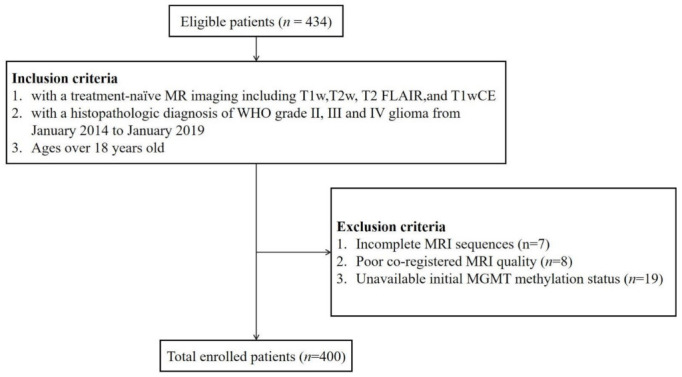
Patient inclusion and exclusion criteria. Abbreviations: T1w, T1-weighted imaging; T2w, T2-weighted imaging; T1wCE, contrast-enhanced T1-weighted imaging; FLAIR, fluid-attenuated inversion recovery.

**Figure 2 cancers-14-04827-f002:**
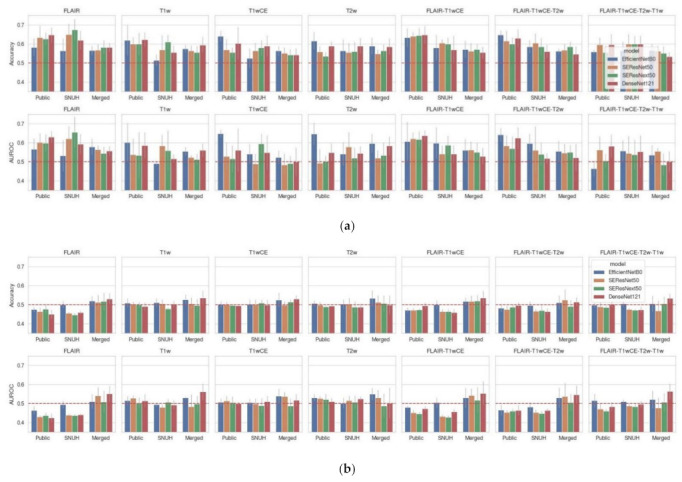
Model performance in the (**a**) validation (or tuning) and (**b**) test sets. For each of the experiments, both accuracy and AUROC are shown to report the model performance in the (**a**) validation (i.e., tuning) and (**b**) test sets. Note that the dashed red lines are the chance level (50%). The horizontal axis is the dataset on which the model was trained/validated (i.e., trained/tuned): “Public” indicates that the model was trained/validated on the BraTS dataset and tested on the SNUH dataset. “SNUH” indicates that the model was trained/validated on the SNUH dataset and tested on the BraTS dataset. “Merged” indicates that the model was trained/validated and tested with a randomly split SNUH + BraTS dataset. Error bars indicate the standard deviation of the metrics. Note that the validation metrics are better than the test metrics because the model training was stopped early according to the high validation accuracy. The red dotted lines indicate the chance level. Abbreviations: AUROC, area under the receiver operating characteristic curve.

**Figure 3 cancers-14-04827-f003:**
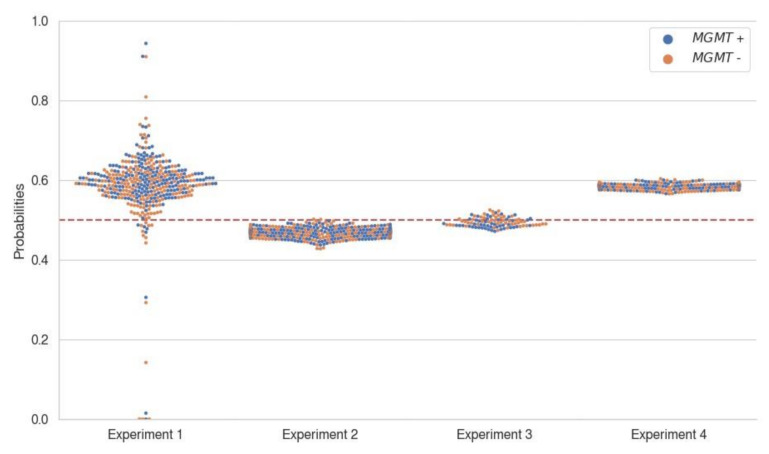
Probability score distribution according to the MGMT labels. The probability scores were predicted from the best model of each experiment (specified in Table 4) and obtained using the test set specific to each experiment. Note that there is no noticeable boundary in the distribution of data points between the groups with high and low probability scores, according to MGMT labels, which are indicated as different colors. MGMT+ (blue) indicates the methylated MGMT promotor group, and MGMT- (orange) indicates the unmethylated MGMT promotor group. The red dotted lines indicate chance level. Abbreviations: MGMT, O6-methylguanine-DNA methyltransferase.

**Table 1 cancers-14-04827-t001:** Patient demographics and genetic information.

	Number of Patients	Age, Mean ± SD (Years)	PFS, Median (95% CI) (Days)	*p* Value
Sex				
Male	240 (60%)	52.6 ± 15.7	327 (287–372)	0.655
Female	160 (40%)	51.9 ± 14.7	362 (301–481)	
MGMT				
Unmethylated	203 (50.8%)	52.6 ± 15.6	396 (328–526)	<0.0001 *
Methylated	197 (49.2%)	52.0 ± 14.9	974 (698–1302)	

Abbreviations: SD = standard deviation; CI = confidence interval; MGMT = O6-methylguanine-DNA methyltransferase; PFS= progression-free survival. * indicates the *p* value for a significant difference in PFS using the log-rank test.

**Table 2 cancers-14-04827-t002:** Comparison of previous prediction models of MGMT methylation.

Previous Study	Dataset	MR Sequence	Input Feature	Model Architecture	Dimension	Diagnostic Performance
Han et al. [8]	TCIA (*n* = 262): Grade IV glioblastoma	T1w, T2w, FLAIR	Raw images	CRNN	2D axial CNN with RNN in slice-direction (z-axis)	Acc 67% (validation), 62% (test), precision (67%), recall (67%)
Sasaki et al. [11]	Osaka International Cancer Institute (*n* = 201): Grade IV glioblastoma	T1w, T2w, FLAIR, T1wCE	Radiomics	Supervised principal component analysis	3D VOI of 1 mm isotropic resampled image	Acc 67% (mean by 10-fold cross-validation)
Levner et al. [10]	Tom Baker Cancer Centre (*n* = 59): Grade IV glioblastoma	T2w, FLAIR, T1wCE	Texture analysis	L1-regularized neural network	2D axial	Acc 87.7%
Drabycz et al. [18]	Tom Baker Cancer Centre (*n* = 59): Grade IV glioblastoma	T2w, FLAIR, T1wCE	Texture analysis	Linear discriminant analysis	2D axial	Acc 71%
Yogananda et al. [12]	TCIA (*n* = 247); Grade II-IV gliomas	T2w	Raw images	3D-DenseUNet	3D patch	Acc 94.7% (mean by 3-fold cross-validation)
Wei et al. [19]	Shanxi Medical University (*n* = 105); Grade II-IV astrocytoma	T1wCE, FLAIR, ADC	Radiomics	Logistic regression	3D VOI	Acc 77% (validation; *n* = 31)
Korfiatis et al. [20]	Mayo Clinic (*n* = 155); Grade IV glioblastoma	T2w, T1wCE	Texture analysis	Support vector machines, random forest classifiers	2D ROI	AUC 0.85

**Table 3 cancers-14-04827-t003:** Comparison of model performance using different models and sequences in validation sets.

Dataset	CNN Architecture	MR Sequence	Metrics ^†^			
			Best AUROC (%)	Accuracy (%)	Precision (%)	Recall (%)
Experiment 1 (Train/valid BraTS, Test SNUH)	EfficientNet-B0	FLAIR-T1wCE-T2w-T1w	46.4 ± 4.7 (39.1–52.0)	55.9 ± 1.2 (54.2–57.6)	55.9 ± 2.8 (53.4–60.7)	83.2 ± 18.7 (54.8–100.0)
Experiment 2 (Train/valid SNUH, Test BraTS)	SEResNeXt50	FLAIR-T1wCE	57.8 ± 8.3 (46.0–67.5)	55.5 ± 5.4 (50.0–62.5)	36.6 ± 34.0 (0.0–71.4)	44.0 ± 49.9 (0.0–100.0)
Experiment 3 (Train/valid SNUH + BraTS, Test SNUH + BraTS)	SEResNet50	T2w	54.9 ± 5.4 (48.8–63.1)	57.1 ± 3.1 (53.9–61.8)	61.6 ± 10.3 (53.2–75.0)	68.7 ± 33.4 (26.1–97.8)

^†^ Metrics were calculated using the validation set for each experiment. The mean and standard deviation were obtained from five different models of the same CNN architectures and MR sequences trained using five different seed numbers. Data are given as the mean ± standard deviation (range).

**Table 4 cancers-14-04827-t004:** Comparison of model performance using different models and sequences in test sets.

Dataset	CNN Architecture	MR Sequence	Metrics ^†^			
			Best AUROC (%)	Accuracy (%)	Precision (%)	Recall (%)
Experiment 1 (Train/valid BraTS, Test SNUH)	EfficientNet-B0	FLAIR-T1wCE-T2w-T1w	51.6 ± 3.8 (47.0–57.2)	49.8 ± 1.3 (48.5–51.5)	49.0 ± 1.8 (45.9–50.5)	80.4 ± 31.5 (25.9–100.0)
Experiment 2 (Train/valid SNUH, Test BraTS)	SEResNeXt50	FLAIR-T1wCE	51.7 ± 7.7 (45.9–64.5)	51.9 ± 3.4 (47.5–55.6)	54.6 ± 5.7 (49.4–64.3)	62.0 ± 36.5 (17.6–94.1)
Experiment 3 (Train/valid SNUH + BraTS, Test SNUH + BraTS)	SEResNet50	T2w	51.5 ± 2.7 (48.5–55.5)	50.4 ± 3.3 (47.3–54.6)	32.6 ± 29.8 (0–55.6)	38.2 ± 43.7 (0–90.2)
Experiment 4 (BraTS winner codes, Test SNUH)	3D-ResNet	T1wCE	56.2	54.8	53.6	59.9

^†^ Metrics were calculated using the test set for each experiment. The mean and standard deviation were obtained from five different models of the same CNN architectures and MR sequences trained using five different seed numbers. Data are given as the mean ± standard deviation (range).

## Data Availability

The datasets generated and/or analysed during the current study are not publicly available due to restrictions but are available from the corresponding author on reasonable request. All codes for model implementation and analysis are uploaded at https://github.com/egyptdj/validating-cnn-mgmt.

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
