# Peer review of "Validation of MRI-Based Models to Predict MGMT Promoter Methylation in Gliomas: BraTS 2021 Radiogenomics Challenge"

_cancers, 2022, doi:10.3390/cancers14194827_

Round 1

Reviewer 1 Report

Authors used multi-institutional database for the model generation and did cross validation with different datasets to prove the robustness of the model given the limitations of available datasets. Although the results obtained from the model are not biologically conclusive and lacks significance, the model itself is robust and can be investigated further for improvement. Manuscript is methodically sound, and limitations are clearly stated.

Author Response

Comment 1:

Authors used multi-institutional database for the model generation and did cross validation with different datasets to prove the robustness of the model given the limitations of available datasets. Although the results obtained from the model are not biologically conclusive and lacks significance, the model itself is robust and can be investigated further for improvement. Manuscript is methodically sound, and limitations are clearly stated.

Response 1:

Thank you for your valuable comments. We agree that our experiments will be able to be used for further investigations.

Reviewer 2 Report

In their manuscript "Validation of CNN-based models to predict MGMT promoter methylation in gliomas: BraTS 2021 Radiogenomics Challenge" Kim and co-workers show that MGMT promoter methylation of gliomas seems not to be predictable from preoperative mulriparametric MRI even when using deep learning strategies.

Although - in principle - a negative result this is important work with a clear conclusion. Moreover, the methods including the datasets, model implementation and deep learning experiments are clearly described and are appropriate for the scientific question.

I have only one major concern that should be addressed in revision. 

The authors included diffuse gliomas WHO grades II - IV according to the 2016 WHO classification.

1. The diagnoses must be re-classified following the guidelines of the actual WHO classification 2021. Otherwise, the manuscript represents historical data.

2. Even following the 2016 WHO classification the authors investigated at least 2 biologically different entities ("tumor types" following the taxonomy of the 2021 WHO classification): IDH-mutant gliomas (astrocytomas/oligodendrogliomas) and IDH-wild-type glioblastomas. Therefore, the authors must include the different diagnoses into their models.

This is particularly important for the calculation of the progression free survival, since it is completely different between IDH-mut and IDH-wt gliomas.

There are some minor shortcomings. The citation in text should be adjusted (actual numbers in line with text). The format of the reference list is different (normal and italic). There are some spell-check errors that should be corrected.

Author Response

Comment 1:

The authors included diffuse gliomas WHO grades II - IV according to the 2016 WHO classification.

  1. The diagnoses must be re-classified following the guidelines of the actual WHO classification 2021. Otherwise, the manuscript represents historical data.
  2. Even following the 2016 WHO classification the authors investigated at least 2 biologically different entities ("tumor types" following the taxonomy of the 2021 WHO classification): IDH-mutant gliomas (astrocytomas/oligodendrogliomas) and IDH-wild-type glioblastomas. Therefore, the authors must include the different diagnoses into their models.

This is particularly important for the calculation of the progression free survival, since it is completely different between IDH-mut and IDH-wt gliomas.

Response 1:

Thank you for your great comments. We followed the 2016 WHO classification, because the BraTS challenge 2021 dataset followed this classification. I totally agree with you that we re-classified the cases and corrected as follows: “Patients with adult-type diffuse gliomas were enrolled in the study: astrocytoma, IDH-mutant (n=63); oligodendroglioma, IDH-mutant, 1p/19q-codeleted (n=33); and glioblastoma, IDH-wildtype (n=304).” from “Patients with diffuse gliomas (i.e., astrocytoma, oligodendroglioma, and glioblastoma) of WHO grade II (n=27), grade III (n=123), and grade IV (n=250) were enrolled in the study.” Moreover, as you pointed out, the glioma cases we have enrolled for the study has at least 2 distinct differential diagnosis with different progression free survival (PFS). However, our model was trained to predict the MGMT methylation, not IDH mutation nor PFS, so we cannot incorporate the differential diagnosis into the model for now, though I think it might be helpful for the improvement of the model performance. We actually performed PFS analysis in the supplementary material, to show the significant difference of PFS between MGMT methylated and unmethylated group. However, this analysis is not changed even if we changed the classification system from 2016 to 2021 WHO CNS tumor classification, and we left the analysis unchanged. Thank you for your helpful comments again.

Comment 2:

There are some minor shortcomings. The citation in text should be adjusted (actual numbers in line with text). The format of the reference list is different (normal and italic). There are some spell-check errors that should be corrected.

Response 2:

Thank you for your comments. We corrected the shortcomings in the citation, reference list, and spelling errors.

Reviewer 3 Report

The authors tested recent CNN architectures via extensive experiments to predict the MGMT methylation status in gliomas. Radiomics is an interesting field of research, whereas currently some applications, and the predictive power of IA tools, are still an open field of investigation. Nevertheless, there are some issues that must be clarified:

- The materials and methods section is, in some part, really hard to read, unclear and in some points really too technical, also considering the journal target and readers. I suggest to slighlty focus more on the manuscript topic rather thatn the technical aspects. For example, they do not properly explain the different radiological characteristics analyzed.

In general, the manuscript sintax is really complex and unclear in some parts.  A major English revision could increase the readers' experience

Author Response

Comment 1:

The materials and methods section is, in some part, really hard to read, unclear and in some points really too technical, also considering the journal target and readers. I suggest to slighlty focus more on the manuscript topic rather thatn the technical aspects. For example, they do not properly explain the different radiological characteristics analyzed.

Response 1:

Thank you for your valuable comments. Our manuscripts were modified to focus on the manuscript topic, and technical details were moved to Supplementary Material. Also, our manuscript included the reason why MGMT methylation status of gliomas may not be predictable with preoperative MRI on radiological aspects as follow

(p.10. line 354) Conflicting results have also been reported that none of the conventional MRI features showed significant differences between the two groups with or without MGMT methyla-tion29-methylation29-35. Moreover, beingas frequent epigenetic changes, MGMT methylation status changes by approximately 15% over the course of treatment36. This might support the idea that methylation of the MGMT promotor is not well reflected by conventional mpMRI well, and conventional imaging findings might be nonspecific except for less edema. Interest-inglyInterestingly, methylation of the MGMT promotor has a low extent of edema, low apparent diffu-siondiffusion coefficient (ADC), and low relative cerebral blood volume (rCBV)8.

(p.11. line 362) The imaging phenotype of gliomas in multiparametric MRI is more largely depend-entdependent on genetic features other than MGMT methylation, such as IDH mutation. There ap-pearappear to be far more clinical and genetic features that more strongly determine the mor-phology of gliomas in conventional MRI, and the methylation status of the MGMT pro-motorpromotor is only one of the “weak” features in determining the imaging phenotype.

Comment 2:

In general, the manuscript sintax is really complex and unclear in some parts. A major English revision could increase the readers' experience.

Response 2:

Thank you for your advice. We improved the English language and style of our manuscript by an English editing service.

Reviewer 4 Report

Dear Authors,

the paper explains several limitation about the approach and methodology. No great comparisons with other models were performed. Considering the value in terms of prospective application there are also several potential issues. The outputs are also exposed with low level of simplicity and attraction for the readers. 

Author Response

Comment 1:

The paper explains several limitation about the approach and methodology. No great comparisons with other models were performed. Considering the value in terms of prospective application there are also several potential issues. The outputs are also exposed with low level of simplicity and attraction for the readers.

Response 1:

Thank you for your valuable comments. We compared a total of 420 models using four different convolutional neural network (CNN) architectures, seven combinations of MR sequences, and three different datasets with five different seed numbers to validate whether CNN-based models may predict MGMT promoter methylation in gliomas. Our results demonstrated that these models are not enough to predict MGMT promoter methylation. The purpose of this study was not to develop but to validate the prediction model, which means that we did not intend to further improve the model to test in the prospective manner. However, we also agreed that considering prospective application of these models are still limited, which requires the further investigation with a larger data sets and other appropriate neural network architectures. These limitations are considered in the last paragraph of the Discussion section as follows.

(p.11. line 375) 1) we cannot conclude that MGMT prediction is “impossible” using conventional mpMRI. However, it was not achievable with a dataset size of nearly 1000 via thorough experiments, including the external validation of 1st place solution of BraTS 2021 challenge; 2) the risk of overfitting may be alleviated by increasing the number of dataset (n=985), but still relatively small dataset using modern “heavy” neural network architectures with large number (i.e., tens of millions) of parameters, which can be used to extract good latent features compared with previous models.

We also changed the following sentence, adding a phrase as follows:

(p.11. line 371) “In summary, using a larger dataset of only conventional MRI sequences cannot be used to significantly improve the diagnostic performance, which suggests that at least additional information is required for improvement, hopefully in the prospective study in the future.”

Round 2

Reviewer 2 Report

In the revised version of the manuscript "Validation of CNN-based models to predict MGMT promoter methylation in gliomas: BraTS 2021 Radiogenomics Challenge" Kim and co-workers addressed the major concern by this reviewer in a way that the authors included the re-classified entities (tumor types) following the 2021 WHO classification. The authors did not include the different diagnoses in their models. This is, unfortunately, rather disappointing. It would be necessary to calculate the models for the two tumor biologies in a separate form.

The argument, that the authors did not calculate the IDH-status is not the point. They performed their analysis on two different tumors. Also, it seems hard to trust the PFS calculation in the supplementary materials. Without the knowledge of the distribution of the different diagnoses the calculation of PFS based on MGMT status doesn‘t make sense. With this way of thinking we could omit the WHO classification and diagnose gliomas only based on their MGMT status. 

Author Response

Response to Reviewer 2 Comments

Point 1: In the revised version of the manuscript "Validation of CNN-based models to predict MGMT promoter methylation in gliomas: BraTS 2021 Radiogenomics Challenge" Kim and co-workers addressed the major concern by this reviewer in a way that the authors included the re-classified entities (tumor types) following the 2021 WHO classification. The authors did not include the different diagnoses in their models. This is, unfortunately, rather disappointing. It would be necessary to calculate the models for the two tumor biologies in a separate form.

Response 1: Thank you for your kind and valuable comments. Sorry for my misunderstanding, and I totally agree with your suggestion to include more specific diagnoses (at least IDH-wt, or IDH-mut) in our MGMT prediction model, because diffuse glioma is a heterogenous disease group. However, I think we should more specifically explained our situtation that the BraTS 2021 challenge did not consider the WHO CNS tumor classification 2021, because the dataset was collected according to the previous classification system. They just collected all WHO grade II-IV diffse gliomas (i.e. glioblastoma, oligodendroglioma, and astrocytoma) as stated in the manuscript, and used them as the training set (n=585). Because there was not enough specific description of BraTS dataset, we added the sentence in "Datasets" section as follows: "The BraTS dataset, one of the largest benchmarks of brain tumor, is a retrospectively collected dataset of brain tumor, and mpMRI scans acquired from multiple institutions under standard clinical conditions. Inclusion criteria comprised pathologically confirmed diagnosis of diffuse gliomas and available MGMT promoter methylation status [9]. ". Therefore, our model, which was also developed and submitted to the BraTS 2021 challenge, was not developed considering the new classification as well. However, because we have the pathologic report including genetic mutations, the revised diagnosis according to the new classification was possible in our hospital dataset. More specifically, we cannot re-trained our model incorporating the new classification, because the BraTS dataset (n=585) do not follow the new classification, and if we cannot use the BraTS dataset as training set, then we lose the novelty of multicenterd large-scale study with external validation. Plesase be aware that even the largest benchmark have not yet collected the diffuse gliomas according to WHO CNS tumor classification 2021, which is not possible for us to do for the current study. To do the re-training, we should develop and validate using only our single center dataset (n=400), which will be a totally different study, which is also the reason that we specified ": BraTS 2021 Radiogenomics Challenge" at the end of the title.

However, of course, following the new classification should be considered for future study, because it’s 2022 already. So, we added the the point you made to the limitation as follows: “3) The prediction model did not consider the IDH status, because the BraTS dataset did not have labels of IDH status for training, which warrants further investigation, because IDH status is crucial in the new WHO CNS tumor classification 2021. For future study, we can incorporate the IDH status to the prediction model for the MGMT methylation in dif-fuse gliomas, because IDH-wildtype and IDH-mutant group shows different tumor biolo-gy, although they are classified as adult-type diffuse gliomas, which might lead to differ-ent diagnostic performance of MGMT methylation.”.

Point 2: The argument, that the authors did not calculate the IDH-status is not the point. They performed their analysis on two different tumors. Also, it seems hard to trust the PFS calculation in the supplementary materials. Without the knowledge of the distribution of the different diagnoses the calculation of PFS based on MGMT status doesn‘t make sense. With this way of thinking we could omit the WHO classification and diagnose gliomas only based on their MGMT status..

Response 2: Thank you again for your kind and valuable comments. As you pointed out, PFS should be different according not only to MGMT methylation, but also to IDH mutation, and should not be analyzed collectively (i.e. diffuse gliomas), because they are, in fact, a disease group in a way. Following your suggestion, we added the PFS analysis and resulting Kaplan-Meier survival curves according to each of the IDH group in Supplementary material. We accordingly corrected the results section as follows: “In subgroup analysis of glioblastoma, IDH-wildtype according to WHO CNS tumor classification 2021, PFS was significantly longer (median, 514 days (95% CI, 438-676 days) vs median, 328 days (95% CI, 285-387 days)) in the methylated group than in the unmethylated group (p=0.0001) (Supplementary Fig. S2); and patients with methylated MGMT showed a HR of 0.51 (95% CI, 0.36-0.72). However, PFS showed no difference between MGMT methylated and unmethylated group in IDH-mutant subgroup: mean, 1949 (95% CI, 1722-2177) vs 1650 (95% CI, 1263-2037) days (p=0.871); HR, 0.91 (95% CI, 0.30-2.76)”.

Reviewer 3 Report

The manuscript is now more clear and focused

Author Response

Thank you for your kind and valuable comments.

Reviewer 4 Report

Dear authors, methodologies as poor and with limits in the generalizations of the insights related to the outcomes. There are some issues during the design specifications and statistical approaches.

Author Response

Thank you for your kind comments. It would be more helpful to specify which part is poor in methodology. We think this study is meaningful in that we validated the performance of MGMT prediction models via extensive experiments, including external validation, using the multicentered large-scale dataset for the first time. 

Round 3

Reviewer 2 Report

Thank you for your comments.